# Hypocalcemia Is a Common Risk Factor for Osteoporosis in Taiwanese Patients with Cushing’s Syndrome

**DOI:** 10.3390/ijerph192316064

**Published:** 2022-11-30

**Authors:** Yung-Nien Chen, Jia-Ruei Tsai, Jung-Fu Chen, Feng-Chih Shen

**Affiliations:** Division of Endocrinology and Metabolism, Department of Internal Medicine, Kaohsiung Chang Gung Memorial Hospital and Chang Gung University College of Medicine, Kaohsiung 833, Taiwan

**Keywords:** Cushing’s syndrome, osteoporosis, heart failure, calcium

## Abstract

Background: Osteoporosis is a cardinal manifestation of Cushing’s syndrome. There is a lack of relevant research on risk factors for osteoporosis among patients with Cushing’s syndrome (CS) in Taiwan. Thus, this study was designed to explore the possible risk factors of osteoporosis. Methods: We gathered patients with a diagnosis of CS between 2001 and 2017 in the Chang Gung Research Database (CGRD). We extracted data including diagnoses and biochemistry from hospital records. The diagnosis of CS was based on ICD-9-CM codes (255.0). Osteoporosis was defined by a T value equal to or less than −2.5 in BMD examination and hypocalcemia was defined as serum calcium concentrations < 8.0 mg/dL. Results: A total of 356 patients with CS who made regular visits to the outpatient department were enrolled in this study. The mean age was 68.6 years, and 74.9% of the patients were female. Of them, 207 patients (58.1%) were diagnosed with osteoporosis. Multivariable logistic regression models indicated that serum calcium level was negatively associated with osteoporosis (OR 0.70, CI 0.54–0.91, *p* < 0.001) after adjustment for age, sex, and other confounding risk factors. In addition, hypocalcemia was associated with heart failure (HF) (OR 2.14, CI 1.02–4.47, *p* < 0.05), stroke (OR 2.58, CI 1.21–5.46, *p* < 0.05) and osteoporosis (OR 3.04, CI 1.24–7.41, *p* < 0.05) in multivariate analysis. Conclusions: Our study found that lower serum calcium levels were common among patients with CS and osteoporosis. Furthermore, CS patients with HF or stroke had high proportion of hypocalcemia. Therefore, these patients must pay more attention to adequate calcium supplementation and undergo the appropriate osteoporosis drug treatment to reduce the risk of subsequent fracture and disability.

## 1. Introduction

Cushing’s syndrome (CS) is a disease that causes a series of metabolic disorders and pathological changes in the whole body due to excessive glucocorticoids in the patient’s body. Excessive glucocorticoid production in patients with CS can originate from exogenous (also known as iatrogenic) or endogenous sources. Studies have found that the most common cause of Cushing’s syndrome is endogenous (up to 65.4%) [1,2]. Endogenous CS is a rare disease with an incidence of about 0.7–2.4 per million people per year [1]. Previous studies have shown that ACTH-dependent Cushing’s syndrome accounts for 80–85% of endogenous CS in adults, of which Cushing’s disease (CD) accounts for 75–80% of all ACTH-dependent CS [1]. Some studies have found, however, that non-ACTH-dependent CS accounts for as high as 56–75% of endogenous CS, and CD accounts for the low rate of about 20–30% in Taiwan [3,4].

Osteoporosis is a cardinal manifestation of Cushing’s syndrome. Endogenous excess of glucocorticoids has a well-documented effect on bone health and is the commonest cause of secondary osteoporosis and bone fracture [5]. Previous research has suggested that the impairment of bone status was estimated in 64–100% of patients with CS. In addition, osteopenia developed in 40–78%, osteoporosis in 22–57%, and bone fractures in 11–76% of patients [6,7,8,9]. The European Registry on Cushing’s Syndrome (ERCUSYN) reported that about 40% of men and 20% of women had vertebral osteoporosis in the two-year follow-up results [6]. Among 80 Italian patients with endogenous CS, up to 76% had vertebral fractures after X-ray examination, but 48% had no specific symptoms [8]. A Danish study of 104 patients with endogenous CS found that they had a 5.4-fold higher risk of low-energy fractures compared to controls [10].

Multifactorial pathogenesis of bone loss in CS has been delineated in previous research, such as hypogonadotropic hypogonadism, a decrease in intestinal calcium absorption and an increase in renal calcium excretion, resulting in secondary hyperparathyroidism [5,11]. Additionally, some studies have demonstrated bone loss in CS was related to suppressed osteoblast and enhanced osteoclast activities [12,13].

At present, there is still a lack of relevant research on the risk of osteoporosis in patients with CS in Taiwan. Identifying possible underlying causes is the most important step in the treatment of secondary osteoporosis; thus, this study aims to explore the possible risk factors and correlation between CS and osteoporosis. The results can be used as a reference for improving the quality of nursing patients with CS and osteoporosis.

## 2. Materials and Methods

### 2.1. Data Source

The Chang Gung Medical Foundation (CGMF) is the largest medical system in Taiwan. Electronic medical records (EMRs) derived from seven Chang Gung Memorial Hospitals (CGMHs) can be used by researchers to provide real-world evidence and improve clinical decision making. The Chang Gung Research Database (CGRD) contains the EMRs of the CGMHs, which can offer extensive clinical details for analysis, such as pathology reports, laboratory values, inpatient and outpatient records and disease category data. More detailed information about the CGRD is available in other studies [14,15]. To ensure data privacy, patient and provider information were encrypted and de-identified. This study was conducted according to the guidelines of the Declaration of Helsinki and Good Clinical Practice and approved by the Institutional Review Board (IRB) of CGMH (approval number 201801066B0). Informed consent was waived according to IRB regulations.

### 2.2. Selection of Patients with Cushing’s Syndrome

This was a retrospective cohort study using registered data in the CGRD from 1 January 2001 to 31 December 2017. First, 4939 patients who had Cushing’s syndrome (CS) were selected. The diagnosis of CS was based on ICD-9-CM codes (255.0), and during the follow-up study period, one of the following criteria needed to be met: >1 inpatient admission with the diagnosis of CS, >3 outpatient visits with a diagnosis of CS, and regular follow-up visits for at least one year. Second, the selected participants who had a report of bone mineral density (BMD) at the femoral neck and lumbar spine by dual energy X-ray absorptiometry (DXA) (Lunar iDXA, GE Healthcare) were recruited. We excluded patients with <1 year of continuous follow-up visits in the outpatient clinic at the CGM and those who did not have complete blood calcium and albumin test data.

### 2.3. Definition of Steoporosis and Hypocalcemia

Osteoporosis was defined by the T value equal to or less than −2.5 measured at either the femoral neck or lumbar spine (L1-L4) in a BMD examination. The serum calcium levels of less than 8.0 mg/dL were defined as hypocalcemia. HF was defined according to ICD-9CM codes (398, 402, and 428) and those with at least three regular follow-up outpatient visits or at least one inpatient admission. The patients’ serum creatinine values and estimated glomerular filtration rate (e-GFR), AST, ALT, and calcium levels were obtained from the clinical laboratory system within 7 days of the BMD examination (index date). Data sources were linked using the unique and permanent CGRD identification number.

### 2.4. Statistical Analysis

Continuous parameters are expressed as the means with standard deviations for normally distributed variables, and those variables that are not normally distributed are presented with mean with interquartile range. The independent *t*-test was used for comparing normally distributed continuous variables and the Mann–Whitney U test for variables not normally distributed between patients with and without osteoporosis or HF. Categorical data are presented as numbers and percentages. The differences between categorical variables were tested by chi-square analysis. Logistic regression analysis was applied based on significant variables by univariate analysis. In addition, those statistically significant factors in univariable analysis were included to identify the independent risk factors for osteoporosis or hypocalcemia by binary logistic multivariable regression analysis. The results of logistic regression analyses are expressed as an OR with a 95% CI. All statistical analyses were performed using SPSS (version 20; SPSS, Inc., Chicago, IL, USA). A *p* value of <0.05 was considered to be significant.

## 3. Results

### 3.1. Study Process Flowchart

Figure 1 shows the flowchart for the selection of the study subjects. Data from 4939 patients with a diagnosis of CS were analyzed. Ultimately, there were 356 patients with CS who had completed a bone mineral density exam.

### 3.2. Different Characteristics and Comorbidities between Patients with and without Osteoporosis

The mean age was 68.6 years, and 74.9% of the patients were female (Table 1). Of them, a total of 207 patients (58.1%) were diagnosed with osteoporosis, and their mean age was 72.2 years, which was older than that of those without osteoporosis. Patients with osteoporosis had a lower BMI (25.3 ± 5.0 vs. 26.6 ± 5.0, *p* = 0.022) and lower serum calcium levels (8.6 ± 0.9 vs. 8.8 ± 0.8, *p* = 0.021) as compared to those without osteoporosis. In addition, among the participants with osteoporosis, the number of patients with heart failure (HF) was higher than that among those without (24.5% vs.14.7%, *p* = 0.024).

HF had a positive association with osteoporosis (OR 1.89, CI 1.09–3.28, *p* < 0.05) in univariate analysis (Table 2). After adjustment for age and sex, however, the significance of correlation disappeared. Serum calcium level was negatively associated with osteoporosis (OR 0.67, CI 0.53–0.84 *p* < 0.001) in univariate analysis, and the correlation remained significant in multivariate analysis after adjustment for age, sex, and other confounding risk factors (OR 0.70, CI 0.54–0.91, *p* < 0.001).

### 3.3. Different Characteristics and Comorbidities between Patients with and without Heart Failure

Because patients with osteoporosis had a higher prevalence of hypocalcemia in our study, we subsequently divided our patients into two groups according to patients with or without hypocalcemia. As shown in Table 3, compared with the patients without hypocalcemia, those with hypocalcemia were older (73.1 vs. 67.9 years, *p* = 0.008) and had lower eGFR (41.9 vs. 70.0 mg/mL/1.73 m^2^, *p* = 0.002) and BMI (24.3 vs. 26.1 kg/m^2^, *p* = 0.027). Furthermore, patients with hypocalcemia had a higher proportion of diabetes (74.5% vs. 54.3%, *p* = 0.011), hypertension (93.6% vs. 74.9%, *p* = 0.003), stroke (40.4% vs. 16.1%, *p* < 0.001), and HF (40.5% vs.17.4%, *p* = 0.001). In multivariate analysis, hypocalcemia was associated with HF (OR 2.14, CI 1.02–4.47, *p* < 0.05), stroke (OR 2.58, CI 1.21–5.46, *p* < 0.05) and osteoporosis (OR 3.04, CI 1.24–7.41, *p* < 0.05) in our study populations after adjustment for age, sex, and other confounding factors (Table 4).

## 4. Discussion

In this study, we demonstrated that serum calcium levels were associated with the occurrence of osteoporosis among patients with CS. In addition, we found that patients with HF, stroke or osteoporosis had a high prevalence of hypocalcemia, suggesting that these patients may require carefully monitoring calcium levels and need adequate calcium supplementation in clinical practice.

The possible mechanism between hypocalcemia and osteoporosis could be excessive glucocorticoids reducing the absorption of calcium in the gastrointestinal tract due to diminished expression of the epithelial Ca^2+^ channel [16], opposition of vitamin D metabolism [17] and inhibition of renal tubule calcium reabsorption, increasing the excretion of calcium in urine [18]. In addition, the excessive glucocorticoids also influence the production and action of other hormones that regulate bone and calcium metabolism such as growth hormone (GH), insulin-like growth factor-I (IGF-I) and gonadotropins [19].

Since calcium deficiency and CS are risk factors for osteoporosis in adults [20], adequate intake of dietary calcium is reasonable advice to all individuals with CS to reduce the development of osteoporosis, and it is a safe and inexpensive way to minimize fracture risk. The National Osteoporosis Foundation (NOF) recommended calcium consumption of 1000 mg/day for men aged 50–70 years and 1200 mg/day of calcium for women older than 51 years and men older than 71 years [21]. The combination of vitamin D and supplemental calcium can lower the risk of fracture, as demonstrated in controlled clinical trials [22]. Furthermore, calcium supplementation has been shown to slightly increase BMD, and a 15% reduced risk of total fractures and a 30% reduced risk of hip fractures were found by a recent meta-analysis from the NOF [23].

Furthermore, we demonstrated that the patients with CS and HF had high proportions of hypocalcemia. The possible mechanism between hypocalcemia and heart failure is thought to be the vital role that calcium plays in myocardial excitation–contraction coupling [24]. In this case, hypocalcemia may contribute to systolic dysfunction of the left ventricle. It was reported in the literature that 98% of the cases present with reduced LVEF [25]. A meta-analysis including three cohort studies with a total of 70,697 participants reported a 17% increase in the overall risk of incident HF in people with osteoporosis compared with those without [26]. In their subgroup analysis, the Asian populations with osteoporosis had a higher risk of incident HF. It was demonstrated that in a cohort study from Taiwan, the cumulative incidence of HF was 2.24% higher in patients with newly diagnosed osteoporosis as compared to osteoporosis-free subjects during a mean follow-up of 7.1 years [27]. In addition, Yu et al. found that osteoporosis had a 1.26-fold increased risk of developing HF among patients with end-stage renal disease [28].

The association between stroke and hypocalcemia is an incidental finding in our study and it could be related to lower vitamin D in patients with stroke. People who develop a stroke may have limited vitamin D production because of reduced exposure to sunlight and diet quality, among other factors [29,30]. A recent study noticed the people who had strokes were more likely to have a severe vitamin D deficiency among 9680 people from the Dutch city of Rotterdam [31]. The measurement of serum levels of vitamin D, however, was not available in our research.

The mean age of patients with CS in this study was 68.6 ± 12.6 years, the mean age of patients diagnosed with osteoporosis was 72.2 ± 11.9 years, 84.9% were female, and the proportion of patients with osteoporosis was 58.1%, which was significantly higher than the general population in Taiwan [32].

The research object of this study was limited to a certain southern medical center in Taiwan, and the number of patients with Cushing’s syndrome was relatively insufficient. Some test items (such as parathyroid hormone or vitamin D concentration) were not available for all patients, which would affect the statistics. In addition, this study was a cross-sectional study, which could not define the causal relationship. In the future, it is hoped that the study population with CS can be expanded to improve the accuracy of the inference of the research results, and a longitudinal study will be warranted to establish the causal relationship between hypocalcemia and osteoporosis.

## 5. Conclusions

Our study found that hypocalcemia was associated with osteoporosis and HF among patients with CS. Therefore, these patients must pay more attention to adequate calcium supplementation and undergo appropriate osteoporosis drug treatment to reduce the risk of subsequent fracture and disability.

## Figures and Tables

**Figure 1 ijerph-19-16064-f001:**
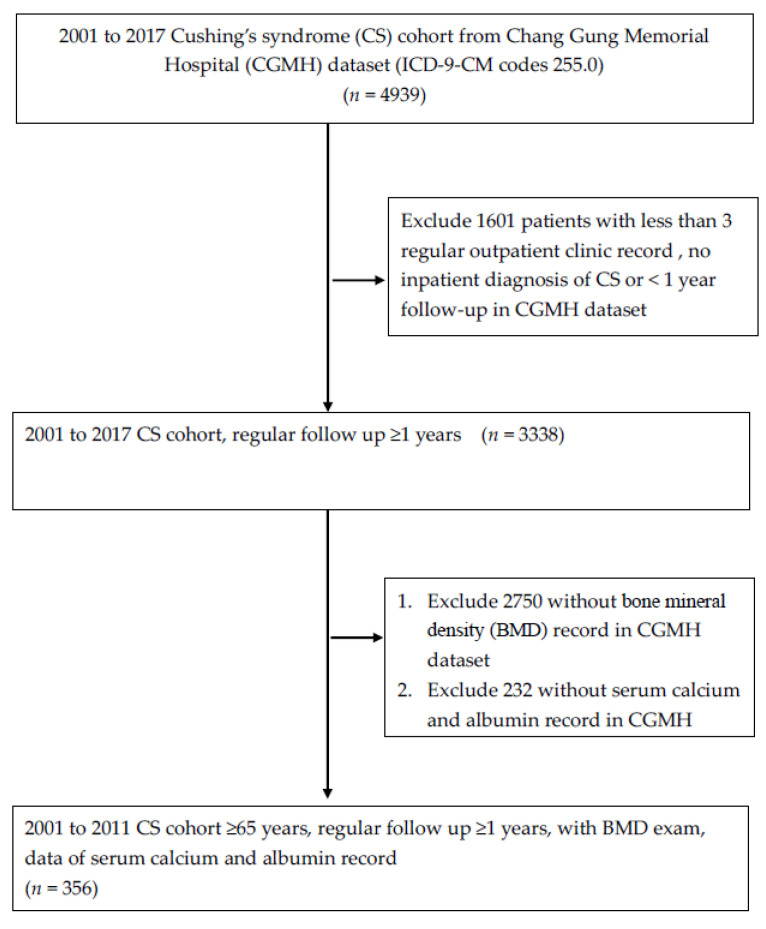
Study process flow chart.

**Table 1 ijerph-19-16064-t001:** The differences in clinical characteristics and comorbidities between those with and without osteoporosis among patients with Cushing’s syndrome.

	All Patients	Without Osteoporosis	With Osteoporosis	*p* Value
*N*	356	149	207	
Age (years)	68.6 ± 12.6	63.9 ± 11.3	72.2 ± 11.9	<0.001
Male	90 (25.1)	55 (36.7)	35 (16.8)	<0.001
Smoking	21 (5.9)	11 (7.3)	10 (4.8)	0.356
Alcohol	6 (1.7)	3 (2.0)	3 (1.4)	0.698
BMI (kg/m^2^)	25.8 ± 5.1	26.6 ± 5.0	25.3 ± 5.0	0.022
Creatinine (mg/dL)	0.91 (0.69, 1.39)	0.88 (0.70, 1.37)	0.95 (0.66, 1.50)	1.000
eGFR (mg/mL/1.73 m^2^)	70.7 ± 44.7	71.7 ± 35.3	70.0 ± 50.4	0.713
AST (U/L)	26.0 (20.0, 35.0)	26.0 (20.0, 35.0)	26.0 (21.0, 34.8)	0.236
ALT (U/L)	21.0 (14.0, 32.0)	22.0 (16.8, 34.2)	20.0 (13.0, 31.0)	0.607
Calcium (mg/dL)	9.1 ± 1.0	9.3 ± 0.8	9.0 ± 1.0	<0.001
Hypocalcemia	47 (13.1)	8 (5.3)	39 (18.8)	<0.001
Diabetes	204 (57.0)	81 (54.0)	123 (59.1)	0.387
Hypertension	277 (77.4)	111 (74.0)	166 (79.8)	0.203
Chronic kidney disease	75 (20.9)	32 (21.3)	43 (20.7)	0.896
Stroke	69 (19.3)	22 (14.7)	47 (22.6)	0.077
Peripheral artery disease	9 (2.5)	1 (0.7)	8 (3.8)	0.086
Coronary artery disease	44 (12.3)	19 (12.7)	25 (12.0)	0.872
Heart failure	73 (20.4)	22 (14.7)	51 (24.5)	0.024
Hyperparathyroidism	46 (12.8)	20 (13.3)	26 (12.5)	0.873
Hyperthyroidism	1 (0.3)	0 (0.0)	1 (0.5)	1.000
Hypogonadism	28 (7.8)	9 (6.0)	19 (9.1)	0.322

Values are displayed as *n* (%), mean ± standard deviation, or mean (interquartile range). BMI, body mass index; AST, Aspartate amino transferase; ALT, Alanine amino transferase.

**Table 2 ijerph-19-16064-t002:** Binary logistic regression models for risk of osteoporosis in patients with Cushing’s syndrome.

	Univariate Analysis	Model 1	Model 2	Model 3
BMI	0.95 (0.91–0.99) *	0.96 (0.92–1.00)	0.96 (0.92–1.00)	0.96 (0.92–1.01)
Calcium	0.67 (0.53–0.84) ^&^	0.70 (0.55–0.90) ^&^	0.68 (0.53–0.89) ^&^	0.70 (0.54–0.91) ^&^
Heart failure	1.89 (1.09–3.28) *	1.37 (0.75–2.50)	1.62 (1.86–3.05)	1.47 (0.77–2.81)

BMI, body mass index. Model 1: adjusted for age and sex. Mode 2: adjusted for model 1 plus smoking, alcohol, diabetes, hypertension, coronary artery disease, chronic kidney disease, stroke, peripheral artery disease and hypogonadism. Model 3: adjusted for mode 2 plus BMI, calcium and heart failure. * *p* < 0.05, ^&^
*p* < 0.01.

**Table 3 ijerph-19-16064-t003:** The differences in clinical characteristics and comorbidities between those with and without hypocalcemia in patients with Cushing’s syndrome.

	Normocalcemia (*n* = 311)	Hypocalcemia (*n* = 47)	*p* Value
Age (years)	67.9 ± 12.6	73.1 ± 11.8	0.008
Male	84 (27.0)	6 (12.8)	0.036
Smoking	20 (6.4)	1 (2.1)	0.334
Alcohol	5 (1.6)	1 (2.1)	0.573
BMI (kg/m^2^)	26.1 ± 5.0	24.3 ± 5.4	0.027
Creatinine (mg/dL)	0.88 (0.69, 1.28)	1.52 (0.85, 2.94)	<0.001
eGFR (mg/mL/1.73 m^2^)	70.0 (48.8, 95.8)	41.9 (15.2, 75.2)	0.002
AST (U/L)	26.0 (21.0, 35.3)	26.0 (19.0, 33.0)	0.188
ALT (U/L)	21.0 (14.0, 33.0)	20.0 (15.0, 29.0)	0.146
Calcium (mg/dL)	9.4 ± 0.8	7.5 ± 0.5	<0.001
Diabetes	169 (54.3)	35 (74.5)	0.011
Hypertension	233 (74.9)	44 (93.6)	0.003
Chronic kidney disease	58 (18.6)	17 (36.2)	0.011
Stroke	50 (16.1)	19 (40.4)	<0.001
Peripheral artery disease	6 (1.9)	3 (6.4)	0.101
Coronary artery disease	35 (11.3)	9 (19.1)	0.150
Heart failure	54 (17.4)	19 (40.5)	0.001
Osteoporosis	169 (54.3)	39 (83.0)	<0.001
Hyperparathyroidism	34 (10.9)	12 (25.5)	0.009
Hyperthyroidism	1 (0.3)	0 (0.0)	1.000
Hypogonadism	27 (8.3)	1 (2.1)	0.150

Values are displayed as *n* (%), mean ± standard deviation, or mean (interquartile range). BMI, body mass index; AST, Aspartate amino transferase; ALT, Alanine amino transferase.

**Table 4 ijerph-19-16064-t004:** Binary logistic regression models for risk of hypocalcemia in patients with Cushing’s syndrome.

	Univariate Analysis	Model 1	Model 2	Model 3
BMI	0.93 (0.97–0.99) *	0.93 (0.87–1.00)	0.94 (0.87–1.01)	0.95 (0.88–1.02)
eGFR	0.99 (0.98–0.99) ^&^	0.99 (0.98–0.99) ^&^	0.99 (0.98–0.99) *	0.99 (0.98–1.00)
Diabetes	2.45 (1.23–4.90) *	2.41 (1.19–4.87) *	2.15 (1.05–4.41) *	1.55 (0.70–3.42)
Hypertension	4.91 (1.48–16.26) ^&^	4.63 (1.39–15.44) *	4.44 (1.32–14.9) *	2.77 (0.77–10.05)
Stroke	3.54 (1.84–6.83) ^&^	3.10 (1.57–6.14) ^&^	3.11 (1.55–6.25) ^&^	2.58 (1.21–5.46) *
Heart failure	3.23 (1.68–6.20) ^&^	2.71 (1.39–5.28) ^&^	2.50 (1.25–5.00) *	2.14 (1.02–4.47) *
Osteoporosis	4.10 (1.85–9.05) ^&^	2.99 (1.29–6.93) ^&^	3.05 (1.31–7.12) *	3.04 (1.24–7.41) *
Hyperparathyroidism	2.79 (1.33–5.89) ^&^	3.04 (1.41–6.58) ^&^	2.83 (1.25–6.38) *	2.00 (0.80–5.03)

BMI, body mass index. Model 1: adjusted for age and sex. Model 2: adjusted for model 1 plus smoking, alcohol, peripheral artery disease, CAD and hypogonadism. Model 3: adjusted for mode 2 plus BMI, eGFR, diabetes, hypertension, stroke, heart failure, osteoporosis and hyperparathyroidism. * *p* < 0.05, ^&^
*p* < 0.01.

## Data Availability

Not applicable.

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
