# Peer review of "Hypocalcemia Is a Common Risk Factor for Osteoporosis in Taiwanese Patients with Cushing’s Syndrome"

_ijerph, 2022, doi:10.3390/ijerph192316064_

Round 1

Reviewer 1 Report

The manuscript "The association between serum calcium levels and osteoporosis in patients with Cushing’s syndrome" by Feng Chih Shen presents a retrospective analysis of patients with Cushing's syndrome (CS) within the Chang Gung Research Database in Taiwan, exploring risk factors for osteoporosis in this patient population. The main result is a statistically significant tendency towards lower serum calcium levels in CS patients with osteoporosis compared against those without osteoporosis. Furthermore, a higher frequency of heart failure was found in this patient group, but notwithstanding correction for confounding variables - in this case most likely age. There are several issues with the manuscript and it fails to convince.

In general, the manuscript needs heavy copy-editing and needs to be revised by a native-speaking professional. Many sentences are odd in structure and some words not fitting. At times it is difficult to follow the train of thought. The use of "we" and "our" seems like a misnomer given there is only one author. If this work is based on a collective effort (and it most certainly is), then the author should have the decency to include those whose hard work made the study possible and qualify for authorship according to established criteria.

The title and abstract need to be improved. As a pars pro toto they illustrate the manuscript's problem of a missing overarching structure and line of thought. The title does not adequately reflect the manuscript's content and key message; neither does the abstract. A suggestion for the title would be: "Hypocalcemia is a common risk factor of osteoporosis in Taiwanese patients with Cushing's syndrome" With some creative effort one might come up with an even more appropriate title. Also: Please add confidence intervals to each instance of odd's ratios, e.g. in the abstract and the body text.

The introduction is a little rough around the edges and lacks a clear path from established knowledge to the research question to be addressed. The known links of CS to osteoporosis could be elaborated in more detail, especially those also considering calcium homeostasis.

The methods have several shortcomings. The description of the data source is confusing and doesn't really explain where the data originally come from: The analysis performed uses data from the Chang Gung Research Database. But how does it get into that database? Are those only patients seen in Chang Gung Memorial Hospital or does it recruit patients from all over the country? The number of 4,939 patients with Cushing's syndrome seems unusually high: If we were to assume a prevalence of 100 patients per million inhabitants (which would be a rather high assumption and higher than published epidemiology data) in a country with 23 million inhabitants, a total of 2,300 patients could be expected. The number in this manuscript is more than twice that. This makes one wonder how diagnosis was established and if stringent criteria were applied. That is not entirely clear. It states that ICD-9-CM code 255.0 (what about 255.3?) was used. That by itself doesn't explain a lot. How did all these patients get to be diagnosed with 255.0? Do we know if CS is persisting or are we talking about patients in remission? Or both?

Finally, the results are derived from a subset of approximately 7% of the Cushing's syndrome base population (cf. selection flow-chart), thus introducing a selection bias (that is not discussed, by the way). There are discrepancies between data given in the text, the tables, and the abstract. This shows a lack of diligence while preparing the manuscript. It is not clear, why heart failure is mentioned. The title and introduction do not cover this. The data on heart failure dilute the manuscript and deflect from the core message. Anyhow, even though the discussion makes some interesting observations on possible links, the fact that correcting for confounding variables eliminates the significance should be an obvious sign that the links is explained by the confounders (i.e. age).

The discussion has some merit, but overall fails to adequately put the main findings into perspective, address limitations properly. The information on heart failure and its link to osteoporosis is interesting, but seems beside the point. The main finding is barely discussed, while heart failure makes up approximately half of the discussion.

A strength of the literature is the regional focus with many data from Taiwan, thus fitted for the study population. However, other relevant papers are not cited. To name a few:

* Meng et al. 1989 "Metabolism of calcium and phosphorus in Cushing syndrome with osteoporosis" (PMID: 2627828)

* Mitchell & Lyles. 1990 "Glucocorticoid-induced osteoporosis: mechanisms for bone loss; evaluation of strategies for prevention" (PMID: 2203846)

* Ziegler & Kasperk. 1998 "Glucocorticoid-induced osteoporosis: prevention and treatment" (PMID: 9618799)

* Tomita. 1998 "Glucocorticoid-induced osteoporosis--mechanisms and preventions" (PMID: 9648484)

* Guo et al. 2018 "Effect of hypercortisolism on bone mineral density and bone metabolism: A potential protective effect of adrenocorticotropic hormone in patients with Cushing’s disease" (PMID: 28851260)

All these shortcomings weigh too heavy, which is why I would recommend rejection of the manuscript.

Author Response

Dear reviewer

Your constructive criticism is greatly appreciated. We have made the following responses to comply with your honorable suggestions

The manuscript "The association between serum calcium levels and osteoporosis in patients with Cushing’s syndrome" by Feng Chih Shen presents a retrospective analysis of patients with Cushing's syndrome (CS) within the Chang Gung Research Database in Taiwan, exploring risk factors for osteoporosis in this patient population. The main result is a statistically significant tendency towards lower serum calcium levels in CS patients with osteoporosis compared against those without osteoporosis. Furthermore, a higher frequency of heart failure was found in this patient group, but notwithstanding correction for confounding variables - in this case most likely age. There are several issues with the manuscript and it fails to convince.

Comment 1: In general, the manuscript needs heavy copy-editing and needs to be revised by a native-speaking professional. Many sentences are odd in structure and some words not fitting. At times it is difficult to follow the train of thought. The use of "we" and "our" seems like a misnomer given there is only one author. If this work is based on a collective effort (and it most certainly is), then the author should have the decency to include those whose hard work made the study possible and qualify for authorship according to established criteria.

Response: Thanks for your comments. We have revised our manuscript by editing service. In addition, the list of authorships was changed to include some of my colleague participating the collection and analysis of data and writing the manuscript.

Comment 2: The title and abstract need to be improved. As a pars pro toto they illustrate the manuscript's problem of a missing overarching structure and line of thought. The title does not adequately reflect the manuscript's content and key message; neither does the abstract. A suggestion for the title would be: "Hypocalcemia is a common risk factor of osteoporosis in Taiwanese patients with Cushing's syndrome" With some creative effort one might come up with an even more appropriate title. Also: Please add confidence intervals to each instance of odd's ratios, e.g. in the abstract and the body text.

Response: Thanks for your constructive comments. We have changed the title as your suggestion and added the confidence intervals to odd’s ratios in the abstract and body text. We also made some changes of the contents of abstract.  

Comment 3: The introduction is a little rough around the edges and lacks a clear path from established knowledge to the research question to be addressed. The known links of CS to osteoporosis could be elaborated in more detail, especially those also considering calcium homeostasis.

 Response: Thanks for your comments. We have added more details discussing the link of CS to osteoporosis in Introduction (page 2, line 53).

Comment 4: The methods have several shortcomings. The description of the data source is confusing and doesn't really explain where the data originally come from: The analysis performed uses data from the Chang Gung Research Database. But how does it get into that database? Are those only patients seen in Chang Gung Memorial Hospital or does it recruit patients from all over the country? The number of 4,939 patients with Cushing's syndrome seems unusually high: If we were to assume a prevalence of 100 patients per million inhabitants (which would be a rather high assumption and higher than published epidemiology data) in a country with 23 million inhabitants, a total of 2,300 patients could be expected. The number in this manuscript is more than twice that. This makes one wonder how diagnosis was established and if stringent criteria were applied. That is not entirely clear. It states that ICD-9-CM code 255.0 (what about 255.3?) was used. That by itself doesn't explain a lot. How did all these patients get to be diagnosed with 255.0? Do we know if CS is persisting or are we talking about patients in remission? Or both?

Response: Thanks for your comments. We have added more details about Chang Gung Research Database in the methods (Page 2, Line 66). The unusually high proportions of patients diagnosed with Cushing’s syndrome (CS) in the screening could be related to high prevalence of Iatrogenic CS in Taiwan [1]. In our clinical practice, it is not uncommon that patients taking steroids or folk medicine containing steroid in early age of Taiwan. The diagnosis code of CS would be added for these patients . Thus, we need to exclude these subjects by using the criteria (>1 inpatient admission with the diagnosis of CS, >3 outpatient visits with a diagnosis of CS, and regular follow-up visits for at least one year) to reduce the selection bias.

[1] Liou TC, Lam HC, Ho LT. Cushing's syndrome: analysis of 188 cases. Taiwan Yi Xue Hui Za Zhi. 1989;88:886-93.

Comment 5: Finally, the results are derived from a subset of approximately 7% of the Cushing's syndrome base population (cf. selection flow-chart), thus introducing a selection bias (that is not discussed, by the way). There are discrepancies between data given in the text, the tables, and the abstract. This shows a lack of diligence while preparing the manuscript. It is not clear, why heart failure is mentioned. The title and introduction do not cover this. The data on heart failure dilute the manuscript and deflect from the core message. Anyhow, even though the discussion makes some interesting observations on possible links, the fact that correcting for confounding variables eliminates the significance should be an obvious sign that the links is explained by the confounders (i.e. age).

Response: Thanks for your instructive comments. We have made some change of our analysis and discussion in manuscript. As your comment, the analysis of patients with or without heart failure is not appropriate and illogical. Thus we withdrew the analysis for heart failure and analyzed with risk factors associated with hypocalcemia (Table 3 and 4 in revision manuscript), since it is related to osteoporosis in patients with CS. We think this change could make the manuscript more logically.

Comments 6: The discussion has some merit, but overall fails to adequately put the main findings into perspective, address limitations properly. The information on heart failure and its link to osteoporosis is interesting, but seems beside the point. The main finding is barely discussed, while heart failure makes up approximately half of the discussion.

A strength of the literature is the regional focus with many data from Taiwan, thus fitted for the study population. However, other relevant papers are not cited. To name a few:

* Meng et al. 1989 "Metabolism of calcium and phosphorus in Cushing syndrome with osteoporosis" (PMID: 2627828)

* Mitchell & Lyles. 1990 "Glucocorticoid-induced osteoporosis: mechanisms for bone loss; evaluation of strategies for prevention" (PMID: 2203846)

* Ziegler & Kasperk. 1998 "Glucocorticoid-induced osteoporosis: prevention and treatment" (PMID: 9618799)

* Tomita. 1998 "Glucocorticoid-induced osteoporosis--mechanisms and preventions" (PMID: 9648484)

* Guo et al. 2018 "Effect of hypercortisolism on bone mineral density and bone metabolism: A potential protective effect of adrenocorticotropic hormone in patients with Cushing’s disease" (PMID: 28851260)

All these shortcomings weigh too heavy, which is why I would recommend rejection of the manuscript

Response: Thanks for your comments. We have added discussion about the correlation between hypocalcemia and osteoporosis in section of discussion (Page 7, Line 179). Also, we added some relevant researches in our references (reference 11, 12 and 13)

Reviewer 2 Report

The manuscript studies the correlation between the reduction of blood calcium and osteoporosis in Cushing’s syndrome patients. Conclusion of the paper is also appropriate considering the findings presented. I find the manuscript interesting and there are a few points to improve it further. 

Glucocorticoids decrease calcium absorption in the intestine partly by two means. It opposes vitamin D action and also reduces the expression of calcium channels inside the duodenum. These possible molecular mechanisms should be added to better explain the association. ‘

The term gender should be replaced with sex to better resonate the biological entity. 

It should be specified in the legend of Tables 1 and 3 that these patients are CS patients. 

When median is used to report age, make sure that it is reported along with median absolute deviation or interquartile range. Otherwise, use mean and standard deviation. 

Figure 1 legend is missing.

Author Response

Dear Reviewer,

Your constructive criticism is greatly appreciated. We have made the following responses to comply with your honorable suggestions

The manuscript studies the correlation between the reduction of blood calcium and osteoporosis in Cushing’s syndrome patients. Conclusion of the paper is also appropriate considering the findings presented. I find the manuscript interesting and there are a few points to improve it further. 

Comment 1: Glucocorticoids decrease calcium absorption in the intestine partly by two means. It opposes vitamin D action and also reduces the expression of calcium channels inside the duodenum. These possible molecular mechanisms should be added to better explain the association.

Response: Thanks for your constructive comments. We have added the association between hypocalcemia and osteoporosis in the part of discussion.

Comment 2: The term gender should be replaced with sex to better resonate the biological entity. 

Response: Thanks for you comment. We have replaced the term sex to gender in our manuscript.

Comment 3: It should be specified in the legend of Tables 1 and 3 that these patients are CS patients. 

Response : We have changed the legend of Table 1 and 3.

Comment 4: When median is used to report age, make sure that it is reported along with median absolute deviation or interquartile range. Otherwise, use mean and standard deviation. 

Response : Thanks for your instructive comments. We have to apology for the inappropriate description of reported age in the manuscript. We have change the expression of reported age in mean and standard deviation because of the age is normally distributed.

Comment 5: Figure 1 legend is missing.

Response: We have added the legend for Figure 1.

Round 2

Reviewer 1 Report

The authors have put much effort into improving the manuscript and it shows! Very good job, it has advanced very much. I think it is now almost fit for publication, but it seems as if there were some typos introduced by the additions. Please cross-check the text for spelling.